# Exploring the lived experience and chronic low back pain beliefs of English-speaking Punjabi and white British people: a qualitative study within the NHS

Gurpreet Singh,[1,2] Christopher Newton,[1] Kieran O'Sullivan,[3,4] Andrew Soundy,[2] Nicola R Heneghan[5]

[1]Musculoskeletal Physiotherapy Department, University Hospitals of Leicester NHS Trust, Leicester, UK
[2]School of Sport, Exercise and Rehabilitation Sciences, University of Birmingham, Birmingham, UK
[3]Department of Clinical Therapies, University of Limerick, Limerick, Ireland
[4]Sports Spine Centre, Aspetar Orthopaedic and Sports Medicine Hospital, Doha, Qatar
[5]Centre of Precision Rehabilitation for Spinal Pain, School of Sport, Exercise and Rehabilitation Sciences, University of Birmingham, Birmingham, UK

**Correspondence to**
Dr Nicola R Heneghan;
n.heneghan@bham.ac.uk

## ABSTRACT

**Introduction** Disabling chronic low back pain (CLBP) is associated with negative beliefs and behaviours, which are influenced by culture, religion and interactions with healthcare practitioners (HCPs). In the UK, HCPs encounter people from different cultures and ethnic backgrounds, with South Asian Indians (including Punjabis) forming the largest ethnic minority group. Better understanding of the beliefs and experiences of ethnic minorities with CLBP might inform effective management.

**Objectives** To explore the CLBP beliefs and experiences of English-speaking Punjabi and white British people living with CLBP, explore how beliefs may influence the lived experience of CLBP and conduct cross-cultural comparisons between the two groups.

**Design** Qualitative study using semistructured interviews set within an interpretive description framework and thematic analysis.

**Setting** A National Health Service hospital physiotherapy department, Leicester, UK.

**Participants** 10 CLBP participants (5 English-speaking Punjabi and 5 white British) purposively recruited from physiotherapy waiting lists.

**Results** Participants from both groups held negative biomedical CLBP beliefs such as the 'spine is weak', experienced unfulfilling interactions with HCPs commonly due to a perceived lack of support and negative psychosocial dimensions of CLBP with most participants catastrophising about their CLBP. Specific findings to Punjabi participants included (1) disruption to cultural-religious well-being, as well as (2) a perceived lack of understanding and empathy regarding their CLBP from the Punjabi community. In contrast to their white British counterparts, Punjabi participants reported initially using passive coping strategies; however, all participants reported a transition towards active coping strategies.

**Conclusion** CLBP beliefs and experiences, irrespective of ethnicity, were primarily biomedically orientated. However, cross-cultural differences included cultural-religious well-being, the community response to CLBP experienced by Punjabi participants and coping styles. These findings might help inform management of people with CLBP.

### Strengths and limitations of this study

► The first study to provide a cross-cultural exploration of chronic low back pain (CLBP) beliefs and experiences of English-speaking Punjabi and white British people living with CLBP.
► Using purposive sampling, 1:1 semistructured interviews were conducted through a cultural lens to explore beliefs and experiences of Punjabi and white British people with CLBP.
► The study findings were data driven and embedded in the participants' voice.
► Participants were all English speaking and were only selected from one geographical location, which may limit the transferability of the findings.
► Member checking was not conducted to validate interview transcripts.

## INTRODUCTION

Chronic low back pain (CLBP) is the primary cause of disability and financial burden to healthcare and society in the UK.[1] Despite increasing resources spent to improve CLBP management, the associated disability continues to rise.[2]

Historically, the impact of CLBP on developing populations was perceived to be minimal but is now understood to be similar to Western populations.[3–8] In the UK, a higher incidence of spinal pain has been reported in South Asians,[9] and a higher prevalence of widespread musculoskeletal pain reported in South Asian Indian Punjabis (22%) compared with Europeans (9%).[10]

Biomedical beliefs about CLBP are common within Western populations and are emergent within developing populations.[3 11 12] Factors thought to influence these beliefs include ethnicity, religion, family and

friends, as well as unhelpful interactions with healthcare practitioners (HCPs).[13 14]

Current CLBP research has largely focused on Western societies with little emphasis on minority ethnic populations; a concern given ethnic migration and cultural diversity is increasing within Western societies. According to the last UK Census (2011),[15] Leicester hosts the largest Indian (referring to themselves as Asian or British Asian) population (30%) of any local authority in England and Wales.

Understanding cultural variations in pain perception, beliefs, expectations and behaviours is important to accurately identify patients' needs and behaviours relative to one's own potentially divergent culture.[16] This may help avoid health inequalities and suboptimal outcomes,[17 18] an important consideration for healthcare policy makers and those responsible for service provision. To tailor management, HCPs might benefit from understanding an individual's beliefs and experiences of CLBP within the cultural context in which they occur.[7 19] Therefore, research is required to understand the management of CLBP within different populations within the UK. Cross-cultural comparative studies using qualitative methodologies may provide in-depth understanding of individual and culture-specific beliefs and experiences of CLBP. However, the authors are not aware that any such comparisons have been made between English-speaking Punjabi and white British CLBP populations.

Briefly, Punjabi people are characterised by migrating from the traditional area of the Punjab region in India or Pakistan or having relatives that had done so. They may speak the Punjabi language and share values, customs and beliefs, identifying with Sikhism, Hinduism or Islam.

Therefore, this study aimed to investigate the beliefs and experiences of people living with CLBP in English-speaking Punjabi and white British populations. The objectives were to explore how these beliefs influence, and impact on, the experience of living with CLBP and identify similarities and differences between the two ethnic groups.

## METHODOLOGY

Using semistructured interviews, this study followed the Consolidated Criteria for Reporting Qualitative studies (online supplementary file 1).[20] Interpretative description (ID) was chosen as this qualitative approach has been specifically developed for healthcare enquiries of a clinical phenomenon, using subjective accounts, for the purpose of informing clinical understanding.[21] Prior theoretical and clinical knowledge is valued as a starting point for research in ID, although this can be challenged and developed as the research progresses.[22]

## Sample

Purposive sampling[23] was employed to recruit white British and Punjabi participants sufficient to enable relevant data to be obtained and analysed.[24] The study setting

was Leicester (UK), which hosts a large white British and Punjabi population.

Eligible participants were white British and English-speaking Punjabi people aged 18–65 years with CLBP of ≥6 months' duration.[25] Individuals with a previous history of surgery for CLBP, diagnosed with a specific or a serious underlying cause of their CLBP (ie, fracture, infection, inflammatory spondyloarthropathy, cancer or nerve root compression)[26 27] or who had previous physiotherapy treatment from the authors (GS and CN) were excluded.

## Recruitment

Potential participants were identified by GS following a general practitioner or consultant referral to a National Health Service physiotherapy department for CLBP between April 2014 and April 2015. Study information was posted to eligible individuals with their physiotherapy appointment letter. Following this, an interview was arranged via telephone, prior to physiotherapy commencing. GS obtained written informed consent preinterview. Recruitment continued until saturation was achieved,[28] where robust common themes were established that included knowledge that could also be applied back to and illustrate the individual cases that were identified.[21] All participants were eligible and included. The sample comprised five white British (two males and three females) and five English-speaking Punjabi (three males and two females) people, with a mean age of 40 years (table 1). All Punjabi participants were English-speaking, third-generation, UK-born citizens.

## Data collection

In-depth semistructured interviews were conducted by GS (British Punjabi male) or CN (white British male), with 11 and 13 years musculoskeletal physiotherapy experience, respectively, with a special interest and a priori knowledge of CLBP. Both authors undertook 3 hours of National Institute of Health Research training on semistructured

| Participant code | CLBP duration (years) | Age (years) | Sex | Ethnicity |
|---|---|---|---|---|
| S1 | 25 | 40 | Male | Punjabi |
| S2 | 2 | 51 | Male | White British |
| S3 | 18 | 35 | Female | White British |
| S4 | 20 | 42 | Female | White British |
| S5 | 2 | 23 | Female | Punjabi |
| S6 | 7 | 37 | Male | Punjabi |
| S7 | 4 | 40 | Male | Punjabi |
| S8 | 1 | 53 | Male | White British |
| S9 | 2 | 49 | Female | White British |
| S10 | 14 | 38 | Female | Punjabi |

**Table 1** Participant characteristics

CLBP, chronic low back pain.

**Table 2** Participant pain, disability and psychosocial risk profile data

| Participant code | NPRS | ODI (%) | SFOQ | Sleep | Anxiety | Depression | Catastrophising | Fear-activity | Fear-work | Employment status |
|---|---|---|---|---|---|---|---|---|---|---|
| S1 | 7 | (Moderate) 28 | (Moderate) 48 | 4 | 5 | 4 | 7 | 7 | 3 | Working |
| S2 | 8 | (Moderate) 38 | (High) 59 | 1 | 6 | 5 | 10 | 5 | 5 | Working |
| S3 | 5 | (Moderate) 24 | (Moderate) 46 | 6 | 0 | 0 | 5 | 10 | 5 | Working |
| S4 | 8 | (Moderate) 30 | (Moderate) 48 | 5 | 7 | 2 | 10 | 2 | 2 | Retired |
| S5 | 3 | (High) 51 | (High) 50 | 7 | 6 | 7 | 5 | 0 | 1 | Off work |
| S6 | 2 | (Low) 14 | (Low) 22 | 2 | 0 | 0 | 3 | 4 | 3 | Working |
| S7 | 5 | (Moderate) 24 | (Moderate) 41 | 6 | 3 | 2 | 8 | 7 | 1 | Working |
| S8 | 2 | (Low) 14 | (Low) 29 | 2 | 3 | 1 | 3 | 8 | 3 | Working |
| S9 | 4 | (Moderate) 40 | (High) 73 | 7 | 8 | 9 | 8 | 9 | 3 | Working |
| S10 | 8 | (High) 50 | (High) 77 | 7 | 1 | 6 | 10 | 10 | 7 | Working |

NPRS, Numeric Pain Rating Scale; ODI, Oswestry Disability Index; SFOQ, Short Form Orebro Musculoskeletal Pain Questionnaire.

interviewing.[29] No prior relationship was established with participants; following the interview, all participants commenced physiotherapy treatment with physiotherapists who were not involved with this study. Interviews took place in a quiet room in the physiotherapy department and lasted between 60 min and 70 min.

A topic guide was informed by contemporary CLBP literature[7 13] and research team expertise. This informed the basis and boundary of focus moving forward to analysis in accordance with interpretive description.[21] Further refinements were made following two pilot interviews with CLBP patients.[30] The topic guide included open-ended questions related to the individual's 'story of their CLBP', their beliefs about causation, management and the future as well as the lived experience of CLBP (related to interaction with HCPs, coping with CLBP and its personal, psychological, social and cultural impact) (online supplementary file 2).

Participants provided demographic data and completed validated questionnaires for pain severity (Numerical Pain Rating Scale was a sub-item score from the Short Form Orebro Musculoskeletal Questionnaire (SFOQ)),[31] functional disability (Oswestry Disability Index)[32] and psychosocial risk profile (including sleep, anxiety, depression, catastrophising, fear-activity and fear-work were subitem scores from the SFOQ)[31] (tables 1 and 2).

### Data analysis

Interviews were audio-recorded and transcribed verbatim by GS, who analysed the data using thematic analysis.[33] GS considered each script repeatedly as a way to immerse himself in the data before coding began.[21] Each transcript was analysed line by line using an iterative model immediately after the first interview. This involved: data sampling, collection and analysis occurring in tandem as an ongoing constant comparative process[34] to facilitate the capture of emergent themes during data collection.[33] This process allowed active engagement and familiarisation with the data. However, some of the terms for beliefs and coping were identified from previous literature.[7 13] These terms represented critical analysis and recontexulisation of knowledge from which the analysis could be shaped.[21] From this, initial themes were generated, and data-driven coding facilitated the development of a thematic table, which was modified as data analysis and interpretation evolved (online supplementary file 3). Crucially, this involved critique by the coauthors (CN, KO, AS and NRH)[35] to enhance rigour and trustworthiness of study findings.[23] GS, CN and AS independently assessed the accuracy and completeness of all the transcripts, ensuring these related to the thematic development and emerging themes; this process was collated as an audit trail (online supplementary file 4). Data collection and analysis was transparent and detailed.

### RESULTS

Five main themes emerged from the interviews: (1) biomedical back pain beliefs, (2) coping with CLBP, (3) the psychological and emotional dimensions of living with CLBP, (4) the social and cultural-religious impact of

CLBP and (5) reflecting on HCP interactions, management experience and expectations of future management. These themes are presented in a compare/contrast style between the ethnic groups. Due to the commonalities between the two groups, the findings presented apply to both groups unless otherwise stated (online supplementary file 5).

### Theme 1: biomedical back pain beliefs
#### Cause of CLBP attributed to physical and structural/anatomical factors

All participants held similar biomedical CLBP beliefs. Common causal beliefs attributed CLBP to physical and structural/anatomical factors. These mainly included bending and lifting strains, for example:

> I basically bent down to pick up a pen or something and it clicked and I couldn't straighten myself up. (S6)

Consistent with these beliefs, in cases where a physical causal mechanism could not be recalled, participants self-diagnosed a structural/anatomical cause for their CLBP. The most frequently expressed labels included: 'slipped disc' (n=5), 'wear and tear' (n=3) and 'trapped nerve' (n=3).

#### Recalling HCPs' biomedical diagnosis and the biomedical beliefs adopted

Most participants recalled a diagnostic label derived from HCPs embedded within the biomedical model, consistent with their own beliefs. Nonetheless, some interpreted this information negatively. Following a consultation with a chiropractor, one participant perceived his back '… was out of place' (S7).

Biomedical CLBP beliefs were influenced by manual-handling training and by participants' occupation. A HCP working in a hospital believed the repetitive nature of manual handling in ward settings to be a cause of his CLBP. Subsequently, participants' adopted the belief their spine needed protecting when bending and lifting:

> I suppose if you lift incorrectly it will cause back pain. I mean we have all had it drummed in to us (manual-handling training), how to bend our knees and how to do all that malarkey. (S2)

#### Vulnerability of the spine

Many viewed their spine as vulnerable, central to function and critical to one's well-being. Given participants had 'one back', which was hard to see, combined with a feeling of weakness, the back was commonly described as 'precious' and, when compared with an ankle sprain, required more protection:

> It just feels as though the spine controls so much of your function in your legs and everything, that if you injure it, it's much more serious than perhaps injuring your ankle. (S2)

Consistent with higher levels of disability, participants' fear-avoidance beliefs suggested they believed pain indicated harm and was a warning signal from the back:

> Your back is trying to tell you something. It's trying to tell you to stop doing whatever you're doing if it's hurting…because you are making it worse. (S3)

#### Future outlook

Participants tried to maintain a positive future outlook; this was mainly based on prior pain experiences and individual personalities.

> I even think now that I will wake up and it will all go away and it maybe will. (S2)

Despite this, there was an overriding feeling of uncertainty:

> I don't want to think about that because I don't know how I'm going to be. (S5)

For some Punjabi participants, religion was expressed as a key part of their lives and interlinked with their positive outlook on pain. One participant described a hand injury whereby his fingers were amputated as 'God's will' and expressed his positive outlook in the context of his religious beliefs:

> Someone up there wanted them so they were gone… my sin plate was wiped clean on that day and we start again. (S1)

### Theme 2: coping with CLBP
#### Active coping strategies

White British participants predominantly demonstrated active coping 'self-help' strategies based on prior CLBP experiences and searching online for knowledge and understanding:

> I went onto NHS choices and typed in sciatica - just that word and a whole load of stuff comes up and there is one big sentence and it says 'keep active…'. (S2)

Among the white British group, exercise strategies such as stretching, yoga and football were used. They commonly shared the narrative 'confronting pain and battling on'. Influenced by a need to avoid interference in one's life, fulfil family and parental duties and avoid burdening others, this often meant enduring pain.

> All I was bothered about was getting things better for him (disabled child) because he literally couldn't do anything. (S4)

#### Reliance on HCPs and biomedical interventions

A lack of understanding and control over pain led some participants to rely on HCPs in an unrelenting search for a diagnostic label, while others sought reassurance via scans:

I wanted to have a scan just to see if there was anything major wrong. (S9)

Both groups expressed varying levels of reliance on medication, ranging from daily use to situations of desperation where they felt 'physically stuck' or to prevent pain intensifying.

In contrast to white British participants, Punjabi participants demonstrated a passive reliance on HCPs to provide 'quick fix' interventions including acupuncture, massage and most frequently manual therapies:

You go there (manual therapist) for a quick fix. (S6)

### Protective and avoidance coping strategies

In terms of bending, lifting and heavy physical tasks, many participants used protective and avoidance strategies. Meanwhile, experiencing exercise-related pain increased fear avoidance around exercise, which promoted resting behaviours. Uncertainty about the cause of pain increased hypervigilance to the threat of pain and adherence to manual handling advice to protect the back reflecting a belief of spinal vulnerability:

I think a bit more…. if I have a task that it would probably involve lifting a box or whatever, I will now consciously think, make sure you bend your knees and you keep your back straight if you're lifting something or whatever. Whereas previously you think you are fine, it doesn't matter, nothing is going to happen …. (S1)

### Coping transition

An interesting difference emerged in the coping trajectories of white British and Punjabi participants. Although it was not clear how this transition took place, all Punjabi participants reported a transition from a passive reliance on HCPs for a 'quick fix' to more active coping strategies such as self-searching the internet, as well as replacing rest with increased physical and social activities in order to resume normal life:

Instead of relax it…. I'd just go back to my normal routine. (S6)

### Theme 3: the psychological and emotional dimensions of CLBP
### Psychological and emotional consequences

Across both groups, a number of negative psychological and emotional dimensions of CLBP were expressed. Those with more disabling CLBP most frequently reported depressed mood, hopelessness, frustration, catastrophising thoughts and uncertainty about pain and lack of control over it.

Depressed mood was most often expressed in relation to the persistence of CLBP and as a consequence of failed interventions, disruption to sleep and engagement in meaningful activities. Some attributed disabling CLBP to weight gain resulting in depressed mood:

Because I put on a lot of weight when I was bed bound that got me really down. (S5)

Hopelessness was attributed to a lack of control over pain and a lack of support from HCPs who, in some cases, painted a pessimistic outlook:

No I mean they just said it's just down to wear and tear basically and you've got to live with it. (S9)

Frustration was repeatedly expressed in relation to a lack of understanding of the cause of pain and the interference with daily life and, in some cases, the desire to exercise and lose weight. Participants with more disabling pain held catastrophic CLBP thoughts. Similar to depressed mood and frustration, this was associated with diagnostic uncertainty and a lack of control over pain, resulting in catastrophising about the future:

I might not walk again. (S5)

### Catastrophic nature of pain flare-ups

Often participants used catastrophic language to portray the unpredictable, fluctuating and uncontrollable disabling impact of flare-ups, for example:

I would say at it's very worst point it feels like a ball of fire. It's debilitating to the point where I want to cut my left-side off. It's awful and the longer the pain continues the worse it seems to spread. (S4)

For some, this entailed a physical and emotional struggle. However, only a few seemed to form a link between their thoughts and CLBP:

When I have got a project or something… I am not thinking about my back at all. I am just cracking on. I'm noticing it more because I have got time on my hands. (S4)

### Threat to self-identity

Cross-cultural comparisons highlighted that CLBP posed a major threat to participants' *'self'* and their identity. Some described negative consequences of CLBP such as low self-esteem, reduced independence and disempowerment. One participant expressed disabling CLBP as a major loss:

Everything is just gone now like dignity, confidence. I had a stick but I would only use that sometimes if I was going out. (S5)

During flare-ups, a small number expressed difficulties carrying out daily functional activities. The greatest impact seemed to be on bending-related tasks such as putting on socks and hoovering. This led to dependency on family members and changing roles in their relationships, which negatively impacted participants' psychological and emotional well-being.

### Theme 4: the social and cultural-religious impact of CLBP
### Threat to family/friend relationships and social life

Mostly expressed by white British participants, the interference of CLBP on family relationships and fulfilling

parental roles was described as 'restrictive' on their spouse/partner, often eliciting negative emotions in family members. One participant identified her coping strategies as a potential cause of frustration:

> My husband also gets frustrated with me. He says 'what you being a legend for, why don't you just sit down and do it later'. I don't know… he still thinks I have OCD. (S4)

CLBP inhibited social interactions across both groups. Social isolation was the consequence for some previously sociable participants with more disabling pain:

> Just walking round town I have to stop and go in a café and have a coffee just to sit down to rest my back. I stay at home now. (S9)

### Work impact

Disrupting work roles, those sitting at work constantly fidgeted and had move in an attempt to control pain. Participants also commented how CLBP impacted on colleagues. Sickness absence was discussed by most with varied views, while some cited having time off work due to a flare-up:

> I did have time off 'cos my back was hurting too much. (S7)

### The impact of CLBP on cultural and religious well-being

CLBP negatively impacted cultural-religious well-being of Punjabi participants, consistently impeding meditation, particularly as this involved sitting cross-legged on a floor for long periods:

> I do sit down crossed legs on the floor when we pray and meditate and maybe that has slight impact on it, you know when it maybe gets tired. (S6)

This impact varied from 'the hips and back locking' to an inability to sit, leaving one participant secluding herself from religious rituals that involved sitting:

> (Referring to sitting in the temple) I'll just go when it's quiet y'know, ermmm do my praying and then come back out and then just go downstairs where there's chairs or… I can just hang around somewhere else…I feel excluded sometimes because y'know people tend to like wanna sit upstairs and I have to like go just downstairs. (S10)

Cultural roles and obligations were impacted. In this context, one Punjabi female described the 'perceived' female cultural role and how CLBP disrupted her ability to carry out household duties including cooking. Viewed as essential to the female role among Punjabis, difficulties with or an inability to carry out these duties had far-reaching consequences such as finding a marriage partner:

> With the Asian culture a girl has to do housework - she has to get prepared for her married life, so she

has to learn how to cook, she has to learn how to do housework, look after her husband and when you can't do that, you're you know not suitable anymore. (S10)

### The response of family, friends and wider community to CLBP

Participants reported experiencing varied responses from friends and family members to their CLBP. These included feeling pushed to seek healthcare to 'get it fixed', feeling supported in some cases and oversupported in others. In contrast, one Punjabi participant experienced very little empathy and support:

> My family don't take me seriously anymore because they're just sick of hearing about it and my friends just don't understand. (S10)

Some Punjabi participants felt stigmatised, with cultural comparisons indicating a perceived lack of empathy and understanding from people within the Punjabi community:

> In other cultures… they tend to be a bit more understanding. (S10)

CLBP advice from Punjabi community members, the self-acclaimed *'back pain experts'*, left one participant frustrated:

> Unfortunately we live in a community that everyone thinks they are a qualified doctor. You know, don't do this, do this sort of a thing. (S1)

### Recalling family experiences of CLBP

Participants recalled back pain experiences of family members with regards to their coping strategies, levels of disability and interactions with HCPs. Some family members reluctantly relied on medication, while others adopted active coping strategies and demonstrated self-efficacy. However, participants did not consistently adopt the coping strategies observed. For example, one Punjabi participant initially used passive interventions (including massage) in contrast to his father:

> He has not let it really impact him to be honest because he still goes to the gym, he still lifts weights, just does everything. Before every now and again his back hurts a little bit. He does what he does - he hasn't changed anything. (S6)

### Theme 5: reflecting on HCP interactions, management experience and expectations of future management

#### Varying quality of therapeutic alliance

Reflecting on previous HCP interactions, many experienced variations in therapeutic alliance, with mostly negative interactions. Strong therapeutic alliance was associated with HCPs providing clear communication. This included a clear explanation about the cause of pain, as well as reassurance, collaborative compassionate care with ongoing support and guidance. This increased

adherence to treatment, built empowerment and trust in their HCPs:

> I've great faith in the physio. (S2)

In contrast, several factors were attributed to weak therapeutic alliance. Some depicted a power struggle, where the HCP was in control and access to investigations such as MRI scans was rejected, while others reported a lack of individualised holistic care. However, most cited HCPs communication as a major problem, particularly not being given time, not being taken seriously, not feeling understood nor listened to. One participant reported feeling disrespected:

> It really did upset me when I went to see a consultant - I felt very belittled by how he approached me. (S9)

Expectations were often unmet, where participants did not receive investigations, a clear diagnosis, a physical examination or manual therapy while enduring unexpected pain flare-ups. A perceived lack of guidance and support from HCPs was also cited, in some cases resulting in feelings of helplessness and low mood.

### Interpreting the HCP explanation

Iatrogenic language used by HCPs was fear-inducing for some participants. Interpretations of HCPs explanations led to a sense of vulnerability around the spine and a need to adopt caution. One participant in his 30s recalled his interpretation of 'wear and tear':

> It makes you think that you've got something permanent and you're basically going to have to live with it. (S6)

However, uncertainty about the actual cause of their pain was most common, partly due to mixed messages conveyed by HCPs and inconclusive radiological investigations.

### Appraising interventions and ability to control CLBP

Reflecting on prior CLBP management, this was embedded within the biomedical model for the majority. Treatment interventions were appraised in relation to their therapeutic effect and the participant's ability to control their pain. Medications, despite being taken for long periods, were deemed largely ineffective by many, as was acupuncture. In addition, a reliance on manual therapies offered minimal long-term effect:

> If I'm honest - at the time it's a psychological plaster. (S4)

### Expectations of future management

Interestingly, when exploring cross-cultural expectations of future management, all participants' sought an individualised, mind–body approach, which involved a physical 'hands on' examination. Many expressed the need for HCPs to possess strong communication skills that include: empathy, active listening skills, providing time, clear communication and to:

> Explain things thoroughly, don't frighten the patient, and just generally be welcoming. (S3)

## DISCUSSION

This is the first study to examine the lived experience and CLBP beliefs of English-speaking Punjabi and white British people. Our findings suggest several between-group similarities among most participants including biomedical back pain beliefs, unfulfilling HCP interactions and negative psychological emotional and social influences of CLBP. Differences included CLBP disrupting Punjabi participants' participation in cultural-religious activities. One Punjabi participant reported CLBP disrupted her 'perceived female role' within the home. Many Punjabi participants also experienced a lack of empathy and understanding from the Punjabi community. While white British participants adopted active coping strategies, all of their Punjabi counterparts initially reported a preference for passive coping strategies, but all reported a transition to active coping strategies.

### Biomedical beliefs

Biomedical CLBP beliefs were common among all participants, their family, friends and particularly the wider Punjabi community. This supports the view that biomedical beliefs may not be exclusive to certain populations, instead reflecting the views of Western society overall.[8 36] Biomedical beliefs conveyed by HCPs were adopted by, or were similar to those already held by, participants, consistent with other studies.[13 37] These beliefs were often associated with negative CLBP information,[38] around bending and lifting, perpetuating beliefs of spinal vulnerability culminating in fear-avoidance beliefs and behaviours.[39] Furthermore, the role of cultural-religious pain beliefs in promoting a positive future outlook has been documented in other cultural groups.[40] However, only one Punjabi participant expressed pain beliefs within a positive cultural-religious context, perhaps reflecting participants' predominantly biomedical beliefs.

### Coping strategies and transition

Illustrated as an ongoing challenge by many participants, coping strategies have been shown to influence the development and persistence of CLBP.[41] Most white British participants at times used active coping strategies such as self-searching for knowledge and exercises, thus demonstrating self-efficacy. A recent qualitative CLBP study[42] reported patients require an explanation and understanding of their CLBP, consistent with our study. Many participants searched relentlessly, primarily via reliance on HCPs for biomedical interventions. This perhaps reflects the importance participants attached to finding a diagnosis that would legitimise their pain.[19] Meanwhile, Punjabi participants' initial reliance on HCPs to

provide passive 'quick fix' interventions and a dependency on family members may highlight their biomedical beliefs, underpinned by a lack of understanding and control over pain resulting in low self-efficacy. These coping strategies may have been influenced by interactions with family, or cultural community members, or the HCP management approach. In support, passive coping strategies have been identified in a previous UK study among a South Asian population with chronic pain.[43] However, acculturation levels were low and perhaps participants lacked knowledge about Western medicine. Other UK studies in South Asian populations have identified a reliance on 'complementary' medications.[44] In contrast, in our study, Punjabi participants pursued more conventional Western medications and interventions. This may reflect greater awareness of, or access to, these treatment options, given Punjabi participants were third generation, UK born and likely well acculturated. A novel finding of our study is that all Punjabi participants reported a transition from passive to active coping strategies.[3 5] This may reflect the limited effects of passive interventions, as well as greater knowledge and understanding of CLBP and the potential benefits of active coping strategies. Alternatively, this group may have perceived the HCP as an authoritarian figure and complied with the HCPs approach to management even if it was not their preference.

### Psychological and emotional dimensions

The impact of CLBP has been found to extend beyond physical domains,[3] with many negative and often life-changing psychological and emotional effects.[19 36 45] Contrary to earlier research,[46] participants did not appear to consider these factors as contributors to CLBP, instead viewing these as secondary effects of CLBP. Supporting this, one study found South Asians were unwilling to recognise the influence of psychological, emotional and social factors on their pain.[43] However, negative beliefs about the control of CLBP and the resulting passive coping often reported by participants may have a mediating influence on depressed mood, pain and disability.[47] Feelings of frustration were common among participants due to a lack of explanation and understanding about CLBP, including inconclusive diagnostic radiological investigations. This may reflect their desire for answers linked with pain legitimisation and validation.[48] Contrary to previous research,[49] perceptions of 'not feeling believed' were not consistent with participants' views nor were feelings of anger and perceived injustice associated with the negative impact of CLBP. Perhaps, these feelings did exist but were not expressed due to fear of being judged or it negatively impacting on physiotherapy. Furthermore, our findings lend support to a study showing catastrophic thoughts were associated with a magnified threat to the *'unpredictable'* and *'fluctuating'* nature of pain flare-ups, excessive worry about pain and a pessimistic view of controlling pain.[50]

### Social dimensions

Similar to other studies, CLBP was highly disabling, described as a *'major loss'* by some. It impacted on many aspects of individuals' lives including their identity, self-esteem and independence, leaving some disempowered.[45 51 52] For many, this meant their lives were 'on hold', a finding consistent with a recent systematic review.[36] Our data support the notion that CLBP impacts meaningful relationships, threatening parental and family duties and for some resulting in social isolation.[3 5 45] With regards to work, although participants demonstrated some avoidance behaviours in the workplace, at the time of interviewing, only one participant was absent from work due to CLBP. This may reflect active coping in relation to work, possibly influenced by financial concerns associated with sickness absence, good work support and job satisfaction[53] or positive HCP advice relating to work.

### Therapeutic alliance and a person-centred approach

Strong therapeutic alliance in the management of CLBP has been associated with greater treatment compliance, improved clinical outcomes[54] and greater levels of patient satisfaction.[55] However, most participants experienced weak therapeutic alliance. Associated with weak therapeutic alliance,[56] the management of CLBP for most was viewed as lacking an individualised and holistic approach. Furthermore, participants predominantly experienced a lack of guidance and support and poor HCP communication.[19] Examination of HCP communication revealed a lack of clear explanation and participants' understanding about pain, instead creating uncertainty for many. Language has been identified as an important facet of effective communication given it is personal and variable, particularly among different ethnic populations. Consistent with findings in other populations,[7] HCPs commonly used fear-inducing language which, in combination with biomedical CLBP beliefs and poor HCP communication, is linked to weak therapeutic alliance and CLBP-related disability.[57] These factors may reflect HCPs' lack of interpersonal skills, particularly specialised communication skills and their overutilisation of biomedical approaches to CLBP management,[58] posing a greater challenge to managing ethnic minority populations. These findings question how HCPs interact with people living with CLBP. HCPs may require training to enhance their communication skills and partnerships with patients.[59 60]

It is also worth noting that participants in our study did not experience inequalities in accessing care or treatment, contrary to findings in other South Asian ethnic populations[61 62] who used participants with low acculturation levels. Participants' experience of HCP interactions highlighted a biomedical approach to CLBP. This may be due to the influence of HCPs' biomedical CLBP beliefs on their clinical management.[13] Interestingly, most participants sought an individualised self-management approach[59 60 63] delivered by empathetic HCPs with effective communication,[64] perhaps more aligned with the biopsychosocial model. This quest, along with Punjabi

participants' transition to seek active coping strategies, demonstrates a desire for self-efficacy, which has been linked with reduced disability.[65] HCPs' biomedical preference for managing CLBP may highlight discordance with the biopsychosocial model advocated by recent National Institute for Health and Care Excellence (NICE) guidelines[66] and has been associated with poor adherence to treatment.[13] Thus, more individualised multidimensional approaches to management,[67] built on effective communication facilitating strong therapeutic alliance[68] and self-management might be needed.

## Cultural differences

To date, qualitative research in the UK has paid little attention to the CLBP experience through a cultural and ethnic lens. This requires consideration, given the degree to which individuals identify with their ethnic group, share beliefs and engage in culture roles, which can influence pain experiences.[40] Consistent with other ethnic minority CLBP studies,[7 11] Punjabi participants experienced a disruption to cultural-religious well-being and endured a negative response from the wider community. In this context, some novel and potentially important themes were identified. As Punjabi participants expressed, religion may be important to people in ethnic minority populations[40]; disruption to sitting-based meditation in some cases resulted in immense frustration and isolation. In some populations, the perceived view of the female role involves housework and preparing for marriage or serving the husband.[3 11] This perception may be similar to those held within the Punjabi community. CLBP undermined the ability to carry out these duties. For one participant, this created uncertainty about finding a marriage partner, and feelings of guilt and burden on other female family members, due to increased workloads placed on them. Our findings support those in other ethnic populations where gender differences exist in the experience of CLBP.[3 5] One novel finding of our study was that Punjabi participants perceived a lack of empathy and understanding from the Punjabi community and for some CLBP was a source of stigmatisation.[59] These factors may reflect cultural attitudes towards people with CLBP within Punjabi communities. Influencing factors may include biomedical beliefs held or limited understanding of CLBP. Alternatively, stoicism and perhaps the communication and meaning of pain may differ among these people, and playing down pain may be more acceptable than gesturing emotion.[69] Stoicism towards CLBP may have existed among community members possibly due to levels of acculturation, personal response to or outlook on CLBP. Other influences may include the participants' relationship with the community members they encountered. Findings of this study illustrate the existence of cultural-religious and gender differences specific to Punjabi participants and highlight the need to consider factors specific to the individual in the management of CLBP.[67] It is noteworthy that while the recent NICE guidelines[66] call for a biopsychosocial approach to CLBP, there is no specific guidance on how to acknowledge or manage sociocultural factors and beliefs. Consequently, it may be challenging for HCPs to provide individualised, culturally sensitive biopsychosocial management for patients with CLBP from different ethnic populations.

## Strengths limitations and implications for future research

One of the study strengths is its relevance to clinical practice. This is the first cross-cultural study to explore CLBP beliefs and experiences in English-speaking Punjabi and white British people living with CLBP. The study findings were data driven and embedded in the participants' voice. Reflexivity was demonstrated throughout with the authors declaring how their ethnicity, novice researcher role, special clinical interest in CLBP and a priori knowledge may have influenced data collection, analysis and interpretation.

Waiting for treatment may have influenced participants' response. Member checking was not conducted to validate interview transcripts due to time and funding. High acculturation levels, over-representation of Punjabi male participants and variations in sociodemographic status may limit the transferability of the findings. Therefore, future research could consider Punjabi populations with low acculturation rates in different geographical regions.

## Implications for practice and policy

This study contributes to existing knowledge by providing HCPs managing CLBP in white British and English-speaking Punjabi people new insights, which could improve CLBP management within these groups. There may be specific training needs for HCPs to better understand the multifactorial nature of CLBP, specifically the individual's beliefs and experiences within their psychosocial and cultural-religious context.[58 70] This, in addition to developing a flexible communication style that facilitates strong therapeutic alliance, may help tailor management within a person-centred approach. Other HCP priorities could include disseminating evidence-based beliefs among patients and the public including ethnic minority populations.[7 47]

## CONCLUSION

CLBP beliefs and experiences similar across both groups were biomedically orientated. CLBP was associated with negative psychological and social consequences. Cross-cultural differences related to the negative impact on cultural-religious aspects of Punjabi participants' lives. Punjabi participants also reported a transition from passive to active CLBP coping strategies and experiencing a lack of empathy from Punjabi community members. HCPs should therefore adopt a culturally sensitive approach to the management of CLBP, which considers individuals' beliefs and experiences.

**Acknowledgements** We would like to thank all participants who took the time to share their stories and experiences.

**Contributors** GS together with CN, KO, AS and NRH were responsible for the conception and design of the study. GS and CN were responsible for data collection. GS was responsible for transcription, leading data analysis and initial drafting of the article. All authors contributed to analysis, interpretation and manuscript development. All authors approved final submitted manuscript.

**Funding** Part funded by National Institute of Health Research

**Competing interests** None declared.

**Patient consent** Obtained.

**Ethics approval** Ethical approval was obtained from NRES Committee, London – Riverside, Reference Number: 14/LO/0510.

**Provenance and peer review** Not commissioned; externally peer reviewed.

**Data sharing statement** No additional data available.

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
