## [Reviewer comments · BMJ Open]

ARTICLE DETAILS

TITLE (PROVISIONAL)	Exploring the Lived Experience And Chronic Low Back Pain Beliefs Of English Speaking Punjabi And White British People: a qualitative study within the NHS
AUTHORS	Singh, Gurpreet; Newton, Christopher; O'Sullivan, Kieran; Soundy, Andrew; Heneghan, Nicola

VERSION 1 – REVIEW

REVIEWER	Maude Laliberté Université de Montréal, Canada
REVIEW RETURNED	04-Nov-2017

GENERAL COMMENTS	Thank you for providing me the opportunity to review such an interesting article! This study clearly contributes to the scientific and professional literature with an innovative perspective on a very prevalent issue. Indeed, the cultural/ethnic diversity in our western societies tends to grow and this plurality should influence the way we provide treatments to ensure sound quality of care. Given that, it is essential for health care professionals in general (and physiotherapists in particular) to better understand how the cultural/ethnic background of their patient can influence their therapeutic relationship or the clinical encounter in general. (With this regard, I would suggest that you read Matthew Hunt, Taking culture seriously: consideration for physiotherapists. Physiotherapy 93: 3, 2007). Given the paucity of research of this topic, the qualitative inquiry was a sound choice; especially as the methodological framework used (Thorne framework) aim at analyzing healthcare phenomenon rather than creating new theory. Here are some specific comments: Page 1. Line 3-7 the way the title is formulated seem to imply that the English speaking Punjabi people are not British.... What about Exploring the Lived Experience And Chronic Low Back Pain Beliefs Of White And English Speaking Punjabi People in Britain? Moreover, in the overall article, the order seem to be different.... I.e. The white people in second and the Punjabi people first.... Either adjusts the manuscript or the title to ensure overall coherence. Abstract : P2 line 15-16 seem to involve that the scope of the article is only a comparison between the 2 ethnic/cultural subgroups of the population, but your article scope is broader (not only aiming at understanding ethnic/cultural minorities but.... Patients in general AND compare to better understand the particularity of ethnic/cultural minorities).
--

Results: Again, your results section is very focused on the difference observed for the Punjabi participants while the purpose of the article was broader and many rich similarities were found between the 2 groups.

Also, some results regarding the Punjabi participants were raised by only 1 participant only and, while this perspective is interesting, it should be interpreted with more caution. Therefore, I feel the result section of the abstract need to be refocused on the general themes that emerged from the overall analysis, rather than solely on the difference between the subgroups.

Introduction p. 4 : The introduction is well constructed and clearly underline the importance of the study, in a context of paucity of the literature, but with a highly prevalent problem. However, while you specifically raised the fact that this study would benefit HCP to better understand their patients, I feel it is also important for manager and health care policy maker. Indeed, if there is any specific obstacle that influence the clinical encounter and create health inequities broadly, these professionals (manager, health policy maker) are in a good position to help addressing them.

In your article, there is the presumption that it is clear for the ready what are the dominant cultural factors of the Punjabi community. I would not assume that and would have a small paragraph on cultural dynamics, traditional family structure, and health care seeking behaviour or religious practices.

Methodology p. 7 : line 3. I am surprised that the authors reached « saturation » within an Interpretative Description framework. Indeed, given the complexity of the issue, and the broad variety of factors that can influence these data (gender, age, socioeconomic status, social health determinants, rural vs city living environment, level of pain or functional limitation, type of work, life habits, level of education, immigration status, etc.) it is unlikely in my perspective that nothing new could be understood from the participants.

Reaching saturation would mean that while doing iterative constant comparison analysis, nothing new emerged from the participant. I feel that while the saturation was unlikely reached, the authors could have had a strong and robust analytic structure. (which is still coherent with Thorne framework that refer to the thoughtful clinician test as an indicator of rigor rather than saturation).

Otherwise, the method is well detailed and rigorous.

Results:

Table 1. I am again surprised that white participants are described as White British (meaning they were all British citizen I assume) but Punjabi people... not. However, naively, it seem to be that there could be difference if someone is a new immigrant, or is a citizen of 2nd-3rd generation... Does this involve that all Punjabi participants are not British citizen? I understand that they are all from the Punjabi community and that they speak English but, additional information on their immigration/citizenship status would add to the richness of the data.

Table 2 and 3. Some of the title do not fitting on one line.... Should be adjusted for esthetical concerns. Moreover, not sure it Table 3 useful or had to the paper..... I would put it in appendix

P12. I have a confidentiality concern for S2 participant..... we know a lot about him and I guess he is easily identifiable (it is a nurse, 51 years old, low back pain x 2 years, living or working in Leicester, UK).... So, maybe removed the fact he is a nurse and change for a «healthcare professional working in an hospital»? Or, the table could include age range rather than a specific number (21-30 years old, 31-40 years old....).

Discussion (p21):

Line 7-23 The first summary section could be more nuanced..... is it all participants? Most participants? One participant?

P25 line 31 would be more careful as only one participant raised this....

I feel the discussion could also document the influence of bias toward chronic pain from therapist and the influence of MRI/X-ray diagnosis (that is wanted by the patients from the interview, but not encouraged by HCP- both for the negative impact it can have on the patient and also in the choosing wisely movement aiming to avoid unnecessary treatments/test). Here are some suggested articles that could be interesting to enhance the discussion section:

- Tait RC, Chibnall JT, Kalauokalani D. Provider judgments of patients in pain: Seeking symptom certainty. *Pain Medicine*. 2009;10(1):11-34.
- Solomon PE, Prkachin KM, Farewell V. Enhancing sensitivity to facial expression of pain. *Pain*. 1997;71(3):279-84.
- Jensen MC, Brant-Zawadzki MN, Obuchowski N, Modic MT, Malkasian D, Ross JS. Magnetic resonance imaging of the lumbar spine in people without back pain. *New England Journal of Medicine*. 1994;331(2):69-73.
- Beattie PF, Meyers SP. Magnetic resonance imaging in low back pain: general principles and clinical issues. *Physical Therapy*. 1998;78(7):738-53.
- Abenhaim L, Rossignol M, Gobeille D, Bonvalot Y, Fines P, Scott S. The prognostic consequences in the making of the initial medical diagnosis of work-related back injuries. *Spine*. 1995;20(7):791.
- Prkachin KM, Solomon P, Hwang T, Mercer SR. Does experience influence judgments of pain behaviour? Evidence from relatives of pain patients and therapists. *Pain Res Manag*. 2001;6(2):105-12.
- Marquie L, Raufaste E, Lauque D, Marine C, Ecoiffier M, Sorum P. Pain rating by patients and physicians: evidence of systematic pain miscalibration. *Pain*. 2003;102(3):289-96.
- Moreover, on a more personal note, given the interest of the research group on chronic pain, I am sharing a video made to illustrate the bias of healthcare professional regarding chronic pain: <https://www.youtube.com/watch?v=RJIw1uaig84> This video has English subtitles. In a master project of Mme Sabrina Morin Chabane (Université de Montréal), this video has been used to help sensitize professional to their bias.

Conclusion p27

Line 11 again, would be more careful as only one participant raised this (perceived female role). Also, you have only 2 female Punjabi participants in total.

General comments :

Patients were waiting for physiotherapy. That can influence their behaviour or their view... Maybe patients with low back pain that are not seeking for physiotherapy have different behaviours.

There is data about the prevalence of low back pain in UK.... Is your sample representative of people with low back pain in UK? If not, it should be mentioned as a limitation for the transferability of the findings to other population. In my opinion, there is some overrepresentation of male participant given that chronic problems are more prevalent in women.

	However, I am not aware of UK data. While the interviewer did not have previous relationship with the participant, were they assigned to be their physiotherapist after the interview? Or were the participants automatically assigned to another therapist? This could also have influenced the participant responses.
--	---

REVIEWER	Khean-Jin Goh University of Malaya, Kuala Lumpur, Malaysia
REVIEW RETURNED	16-Nov-2017

GENERAL COMMENTS	This is an interesting topic comparing the experience of CLBP patients of different ethnicity but from a similar geographical area in the UK. I have several comments.  1. Although using a qualitative research methodology, I am uncertain if 5 subjects from each ethnic group is sufficient to develop common themes from their interviews and to reach the conclusions suggested in the paper. 2. Furthermore, levels of acculturation of ethnic Punjabi subjects may differ according to their ages, younger subjects may be more assimilated to British culture and perceptions. 2. Did the authors interview subjects of their own ethnicity? Would the ethnicity of their interviewer affect subjects' responses to the questions? 2. How can the authors explain the initial passive coping responses of Punjabi patients? Could this be due to a cultural perspective of the dominant role of the HCP and that he/she must follow exactly what has been prescribed/advised? Would it be important then for the HCP to encourage the patient be more active in the management of their CLBP? 3. Socio-cultural differences were concluded based on individual subjects' response to being unable to meditate sitting cross-legged and inability to carry out housework. I think this reflects the difficulty posed by CLBP in perform individual activities and am unsure if one can generalise these findings to the whole ethnic community. I think it emphasises the need to individualise care for these patients regardless of ethnicity 3. The authors also suggested that the Punjabi community are culturally less empathic, downplay pain symptoms and expect that patients with CLBP to bear with their pain. On the other hand, epidemiological studies show a high proportion of South Asian patients with musculoskeletal pain. How can this be explained when contrasted with the assumption of Asians being more stoic?
---

VERSION 1 – AUTHOR RESPONSE

Reviewer: 1

Reviewer Name: Maude Laliberté

Institution and Country: Université de Montréal, Canada

Please state any competing interests or state 'None declared': None declared

Please leave your comments for the authors below

For comments, see the document below.

Comment: Thank you for providing me the opportunity to review such an interesting article! This study clearly contributes to the scientific and professional literature with an innovative perspective on a very prevalent issue. Indeed, the cultural/ethnic diversity in our western societies tends to grow and this plurality should influence the way we provide treatments to ensure sound quality of care. Given that, it is essential for health care professionals in general (and physiotherapists in particular) to better understand how the cultural/ethnic background of their patient can influence their therapeutic relationship or the clinical encounter in general. (With this regard, I would suggest that you read Matthew Hunt, Taking culture seriously: consideration for physiotherapists. *Physiotherapy* 93: 3, 2007). Given the paucity of research of this topic, the qualitative inquiry was a sound choice; especially as the methodological framework used (Thorne framework) aim at analyzing healthcare phenomenon rather than creating new theory.

Authors response: Thank you for your comments and for sharing another useful reference.

Here are some specific comments:

Page 1. Line 3-7 the way the title is formulated seem to imply that the English speaking Punjabi people are not British.... What about Exploring the Lived Experience And Chronic Low Back Pain Beliefs Of White And English Speaking Punjabi People in Britain? Moreover, in the overall article, the order seems to be different.... I.e. The white people in second and the Punjabi people first.... Either adjusts the manuscript or the title to ensure overall coherence.

Authors response: We have amended the title; please see track changes on page 1.

Abstract: P2 line 15-16 seem to involve that the scope of the article is only a comparison between the 2 ethnic/cultural subgroups of the population, but your article scope is broader (not only aiming at understanding ethnic/cultural minorities but.... Patients in general AND compare to better understand the particularity of ethnic/cultural minorities).

Authors response: Thank you we have revised this section to better reflect the scope of the study and our planned objectives.

Results: Again, your results section is very focused on the difference observed for the Punjabi participants while the purpose of the article was broader and many rich similarities were found between the 2 groups.

Also, some results regarding the Punjabi participants were raised by only 1 participant only and, while this perspective is interesting, it should be interpreted with more caution. Therefore, I feel the result section of the abstract need to be refocused on the general themes that emerged from the overall analysis, rather than solely on the difference between the subgroups.

Authors response: Thank you we have made the changes to reflect your comments, please see track changes in the results section of the abstract on page 2.

Introduction p.4 : The introduction is well constructed and clearly underline the importance of the study, in a context of paucity of the literature, but with a highly prevalent problem. However, while you specifically raised the fact that this study would benefit HCP to better understand their patients, I feel it is also important for manager and health care policy maker. Indeed, if there is any specific obstacle that influence the clinical encounter and create health inequities broadly, these professionals (manager, health policy maker) are in a good position to help addressing them. In your article, there is the presumption that it is clear for the ready what are the dominant cultural factors of the Punjabi community. I would not assume that and would have a small paragraph on cultural dynamics, traditional family structure, and health care seeking behaviour or religious practices.

Authors response: Thank you for this useful suggestion; we have made changes to reflect your comments, please see track changes in the introduction section on page 4-5. We have also added a small paragraph on Punjabi culture; see track changes in the introduction section on page 5. We hope this addresses your comments.

Methodology p. 7: line 3. I am surprised that the authors reached « saturation » within an Interpretative Description framework. Indeed, given the complexity of the issue, and the broad variety of factors that can influence these data (gender, age, socioeconomic status, social health determinants, rural vs city living environment, level of pain or functional limitation, type of work, life habits, level of education, immigration status, etc.) it is unlikely in my perspective that nothing new could be understood from the participants. Reaching saturation would mean that while doing iterative constant comparison analysis, nothing new emerged from the participant. I feel that while the saturation was unlikely reached, the authors could have had a strong and robust analytic structure. (which is still coherent with Thorne framework that refer to the thoughtful clinician test as an indicator of rigor rather than saturation). Otherwise, the method is well detailed and rigorous.

Authors response: Thank you for your comments. We have made the changes to reflect your comments, please see track changes in the methodology section on page 6-7.

Results:

Table 1. I am again surprised that white participants are described as White British (meaning they were all British citizen I assume) but Punjabi people... not. However, naively, it seem to be that there could be difference if someone is a new immigrant, or is a citizen of 2nd-3rd generation... Does this involve that all Punjabi participants are not British citizen? I understand that they are all from the Punjabi community and that they speak English but, additional information on their immigration/citizenship status would add to the richness of the data.

Authors response: all Punjabi participants were English speaking, third generation UK born citizens. We have added this in the methodology section, please see track changes on pages 7.

Table 2 and 3. Some of the title do not fitting on one line.... Should be adjusted for esthetical concerns. Moreover, not sure if Table 3 useful or had to the paper..... I would put it in appendix

Authors response: Thank you we have made the changes suggested to Table 2 and 3, please see the tracked changes on page 9 and supplementary file 3.

P12. I have a confidentiality concern for S2 participant..... we know a lot about him and I guess he is easily identifiable (it is a nurse, 51 years old, low back pain x 2 years, living or working in Leicester, UK)... So, maybe removed the fact he is a nurse and change for a «healthcare professional working in an hospital»? Or, the table could include age range rather than a specific number (21-30 years old, 31-40 years old....).

Authors response: Thank you for this. We acknowledge your concerns and have removed 'nurse' and replaced it with 'HCP working in a hospital', please see tracked changes on page 11.

Discussion (p21):

Line 7-23 The first summary section could be more nuanced..... is it all participants? Most participants? One participant?

Authors response: Thank you, in light of your comments we have revised this section to: 'This is the first study to examine the lived experience and CLBP beliefs of English speaking Punjabi and White British people. Our findings suggest several between-group similarities amongst most participants including biomedical back pain beliefs, unfulfilling HCP interactions and negative psychological emotional and social influences of CLBP. Differences included CLBP disrupting Punjabi participants' participation in cultural-religious activities, One Punjabi participant reported CLBP disrupted her 'perceived female role' within the home. Many Punjabi participants also experienced a lack of empathy and understanding from the Punjabi community. Whilst White British participants adopted active coping strategies, all of their Punjabi counterparts initially reported a preference for passive coping strategies, but all reported a transition to active coping strategies.' Please see track changes on page 20.

P25 line 31 would be more careful as only one participant raised this....

Authors response: we have made a change to the sentence 'This perception appears similar to those held within the Punjabi community.' Replacing 'appears' with 'may be'. Please see tracked changes on page 24.

I feel the discussion could also document the influence of bias toward chronic pain from therapist and the influence of MRI/X-ray diagnosis (that is wanted by the patients from the interview, but not encouraged by HCP- both for the negative impact it can have on the patient and also in the choosing wisely movement aiming to avoid unnecessary treatments/test). Here are some suggested articles that could be interesting to enhance the discussion section:

- Tait RC, Chibnall JT, Kalauokalani D. Provider judgments of patients in pain: Seeking symptom certainty. *Pain Medicine*. 2009;10(1):11-34.
- Solomon PE, Prkachin KM, Farewell V. Enhancing sensitivity to facial expression of pain. *Pain*. 1997;71(3):279-84.
- Jensen MC, Brant-Zawadzki MN, Obuchowski N, Modic MT, Malkasian D, Ross JS. Magnetic resonance imaging of the lumbar spine in people without back pain. *New England Journal of Medicine*. 1994;331(2):69-73.
- Beattie PF, Meyers SP. Magnetic resonance imaging in low back pain: general principles and clinical issues. *Physical Therapy*. 1998;78(7):738-53.
- Abenhaim L, Rossignol M, Gobeille D, Bonvalot Y, Fines P, Scott S. The prognostic consequences in the making of the initial medical diagnosis of work-related back injuries. *Spine*. 1995;20(7):791.
- Prkachin KM, Solomon P, Hwang T, Mercer SR. Does experience influence judgments of pain behaviour? Evidence from relatives of pain patients and therapists. *Pain Res Manag*. 2001;6(2):105-12.
- Marquie L, Raufaste E, Lauque D, Marine C, Ecoiffier M, Sorum P. Pain rating by patients and physicians: evidence of systematic pain miscalibration. *Pain*. 2003;102(3):289-96.

• Moreover, on a more personal note, given the interest of the research group on chronic pain, I am sharing a video made to illustrate the bias of healthcare professional regarding chronic pain: <https://www.youtube.com/watch?v=RJlw1uaig84> This video has English subtitles. In a master project of Mme Sabrina Morin Chabane (Université de Montréal), this video has been used to help sensitize professional to their bias.

Authors response: Thank you for your comments, suggestions and references. Our data did not show any consistent themes around 'participants wanting imaging and HCPs not being so keen to image.' However, a growing body of evidence highlights imaging and decisions to image are problematic. This is representative of unhelpful beliefs, which are aligned with our findings.

Conclusion p27

Line 11 again, would be more careful as only one participant raised this (perceived female role). Also, you have only 2 female Punjabi participants in total.

Authors response: we have replaced the sentence 'Cross-cultural differences related to the negative impact on cultural-religious aspects of Punjabi participants' lives particularly the perceived female role.', with 'Cross-cultural differences related to the negative impact on cultural-religious aspects of Punjabi participants' lives' please see track changes on page 26. We hope this addresses your comments.

Thank you, we agree we had a sample of only 2 female Punjabi participants; we have amended the limitations section to reflect this, please see track changes on page 25.

General comments:

Patients were waiting for physiotherapy. That can influence their behaviour or their view.... Maybe patients with low back pain that are not seeking for physiotherapy have different behaviours. There is data about the prevalence of low back pain in UK.... Is your sample representative of people with low back pain in UK? If not, it should be mentioned as a limitation for the transferability of the findings to other population. In my opinion, there is some overrepresentation of male participant given that chronic problems are more prevalent in women. However, I am not aware of UK data. While the interviewer did not have previous relationship with the participant, were they assigned to be their physiotherapist after the interview? Or were the participants automatically assigned to another therapist? This could also have influenced the participant responses.

Authors response: Thank you, we agree, waiting for treatment or not could influence participants' response during interview.

Yes, this data (see Table 2) is representative of UK low back pain populations.

We agree there is an over-representation of males in our sample.

Following their interview all participants started physiotherapy treatment with physiotherapists that were not involved with our study. We have taken these valid comments on board and revised the methodology and limitation sections to reflect these, please see track changes on page 7 and 25.

We thank you for reviewing our article; your comments and suggestions have been very useful.

Reviewer: 2

Reviewer Name: Khean-Jin Goh

Institution and Country: University of Malaya, Kuala Lumpur, Malaysia

Please state any competing interests or state 'None declared': None declared

Please leave your comments for the authors below

This is an interesting topic comparing the experience of CLBP patients of different ethnicity but from a similar geographical area in the UK. I have several comments.

1. Although using a qualitative research methodology, I am uncertain if 5 subjects from each ethnic group is sufficient to develop common themes from their interviews and to reach the conclusions suggested in the paper.

Authors response: Thank you we have amended the methodology section to reflect your comments. Please see the tracked changes on page 6-8.

2. Furthermore, levels of acculturation of ethnic Punjabi subjects may differ according to their ages, younger subjects may be more assimilated to British culture and perceptions.

Authors response: Thank you, we agree levels of acculturation may be an influencing factor. We have amended the discussion section to reflect your comments. Please see the tracked changes on page 21.

3. Did the authors interview subjects of their own ethnicity? Would the ethnicity of their interviewer affect subjects' responses to the questions?

Authors response: Thank you, yes, GS and CN interviewed both Punjabi and White-British participants.

We agree interviewer ethnicity could have influenced participants' responses, a consideration discussed in the methodology section. In addition, we have amended the limitations section to reflect your comments. Please see the tracked changes on page 25.

4. How can the authors explain the initial passive coping responses of Punjabi patients? Could this be due to a cultural perspective of the dominant role of the HCP and that he/she must follow exactly what has been prescribed/advised? Would it be important then for the HCP to encourage the patient be more active in the management of their CLBP?

Authors response: Thank you for your comments here. Within the discussion we have explored how the initial passive coping responses of Punjabi participants may be indicative of their biomedical beliefs, underpinned by a lack of understanding and control over pain resulting in low self-efficacy. To reflect your comments we have added 'These coping strategies may have been influenced by interactions with family, or cultural community members, or the HCP management approach.' We have also added 'perceived the HCP as an authoritarian figure'. Please see the tracked changes on page 21.

We agree, and have already discussed the need to promote self-management of CLBP in the discussion section, please see page 24, 'Thus more individualised multi-dimensional approaches to management,⁶⁶ built on effective communication facilitating strong therapeutic alliance⁶⁷ and self-management might be needed. '

5. Socio-cultural differences were concluded based on individual subjects' response to being unable to meditate sitting cross-legged and inability to carry out housework. I think this reflects the difficulty posed by CLBP in perform individual activities and am unsure if one can generalise these findings to the whole ethnic community. I think it emphasises the need to individualise care for these patients regardless of ethnicity

Authors response: Thank you, yes whilst it is difficult to generalise these findings to the whole ethnic community, it may highlight the need to consider factors specific to the individual in the management of CLBP. We have amended the discussion section to reflect your comments. Please see the tracked changes on page 25.

6. The authors also suggested that the Punjabi community are culturally less empathic, downplay pain symptoms and expect that patients with CLBP to bear with their pain. On the other hand, epidemiological studies show a high proportion of South Asian patients with musculoskeletal pain. How can this be explained when contrasted with the assumption of Asians being more stoic?

Authors response: Thank you for this very interesting question. Thank you for this very interesting question. We acknowledge our data may not be generalizable to the wider South Asian communities or indeed wider Punjabi communities. It is also noteworthy the epidemiological studies tended to include people with low acculturation levels.

Stoicism towards CLBP may have existed amongst cultural community members due to levels of acculturation, back pain beliefs held, personal response to, or outlook on CLBP. Other influences may include the participants' relationship with community members. In addition, the communication and meaning of pain may differ amongst these community members compared with those in the epidemiological studies.

Further investigation would be required to assess the generalizability of our findings. We have amended the discussion section to reflect your comments. Please see the tracked changes on page 25.

We thank you for reviewing our article and providing very useful comments.

VERSION 2 – REVIEW

REVIEWER	Maude Laliberté Université de Montréal, Canada
REVIEW RETURNED	25-Dec-2017

GENERAL COMMENTS	I revised the new version of the manuscript and the author responses to the comments. The reviewer comments have been well integrated and I do not have any further comments.
---

REVIEWER	Khean-Jin Goh University of Malaya, Kuala Lumpur, Malaysia
REVIEW RETURNED	20-Dec-2017

GENERAL COMMENTS	Thank you. I have no further comments to add
--